# Brain Glucose Transporters: Role in Pathogenesis and Potential Targets for the Treatment of Alzheimer’s Disease

**DOI:** 10.3390/ijms22158142

**Published:** 2021-07-29

**Authors:** Leszek Szablewski

**Affiliations:** Chair and Department of General Biology and Parasitology, Medical University of Warsaw, 5 Chalubinskiego Str., 02-004 Warsaw, Poland; leszek.szablewski@wum.edu.pl; Tel.: +48-22-621-26-07

**Keywords:** Alzheimer’s disease, glucose transporters, hypometabolism, therapy of Alzheimer’s disease

## Abstract

The most common cause of dementia, especially in elderly people, is Alzheimer’s disease (AD), with aging as its main risk factor. AD is a multifactorial neurodegenerative disease. There are several factors increasing the risk of AD development. One of the main features of Alzheimer’s disease is impairment of brain energy. Hypometabolism caused by decreased glucose uptake is observed in specific areas of the AD-affected brain. Therefore, glucose hypometabolism and energy deficit are hallmarks of AD. There are several hypotheses that explain the role of glucose hypometabolism in AD, but data available on this subject are poor. Reduced transport of glucose into neurons may be related to decreased expression of glucose transporters in neurons and glia. On the other hand, glucose transporters may play a role as potential targets for the treatment of AD. Compounds such as antidiabetic drugs, agonists of SGLT1, insulin, siRNA and liposomes are suggested as therapeutics. Nevertheless, the suggested targets of therapy need further investigations.

## 1. Introduction

The most common cause of dementia, especially in the elderly population, is Alzheimer’s disease (AD). According to the WHO, 35.6 million people have dementia worldwide, and 65.7 million will have dementia in 2030 [1]. Aging is the main risk factor of AD development.

Alzheimer’s disease is a multifactorial neurodegenerative disease. There are two types of AD. Type 1 Alzheimer’s disease is early-onset AD, observed in 5–10% of patients [2,3]. Early-onset AD is related to genetic abnormalities [4]. The other type of AD is sporadic or late-onset AD. This form is observed in 90–95% of patients [2]. The pathology observed in AD is associated with the accumulation of neuritic extracellular amyloid plaques, so-called senile plaques (SPs), in the brain. They are formed by amyloid beta-peptides (AβPs), which are fragments of beta-amyloid precursor protein (APP), and cytosolic neurofibrillary tangles (NFTs). NFTs are aggregates formed due to hyperphosphorylation of cytosolic tau protein. The other pathologies observed in AD are dystrophic neuritis and neurophil threads [5,6]. The classical hypothesis ascribes the pathogenesis of Alzheimer’s diseases to the formation of amyloid beta-peptides oligomers, which form senile plaques and induce neuronal injury, thereby promoting formation of cytosolic neurofibrillary tangles [6]. In early-onset AD, the increased synthesis and accumulation of amyloid beta precursor protein may be an effect of mutations in genes encoding APP, presenilin 1 (PS1) and presenilin 2 (PS2) or inheritance of the Apolipoprotein E e4 (Apo-e4) allele [7].

Impairment of brain energy is one of the main features of Alzheimer’s disease. Decreased glucose uptake and metabolism were observed in specific areas of an AD-affected brain. Production of ATP from glucose metabolism in sporadic AD is decreased by 50%, and this tendency is continued throughout the progression of the disease, causing an approximate 20% deficit [2,8]. Glucose hypometabolism and energy deficit are hallmarks of AD. There are several hypotheses on the role of glucose hypometabolism in Alzheimer’s disease, but data available on this subject are poor.

## 2. Glucose Metabolism in the Brain

A major source of energy for human brain metabolism is glucose; other organs can utilize other sources of energy, such as fatty acids. Neurons in the brain cannot synthesize and store glucose. Although glucose is the sole source of energy for the brain, ketone bodies can be used as well, but only as a last resort [9]. Neurons and glia utilize over 99% of ketone bodies that pass across the plasma membrane. A continuous supply of glucose is necessary for neuronal function; therefore, brain neurons are fully dependent on glucose import.

### 2.1. Human Glucose Transporters

The cell membrane separates the inner environment from the outer environment. It is made up of a lipid bilayer and is selectively permeable to the majority of molecules. Glucose must cross the blood–brain barrier (BBB) and plasma membranes of neurons and glial cells in the central nervous system (CNS). Glucose has a hydrophilic nature, whereas the lipid bilayer is lipophilic; hence, glucose cannot penetrate the lipid bilayer. Therefore, glucose requires specific carrier proteins to be transported across the cell membranes. Carrier proteins, or transporters, are integral membrane proteins involved in the movement of molecules across the plasma membrane.

There are three families of genes encoding glucose transporters in humans: *SLC2A, SLC5A* and *SLC50A* [10]. Although these carrier proteins are called “glucose transporters,” they can transport also other molecules, such as galactose, fructose, mannose, vitamins, inositols, Na^+^, I^+^ and so on. They may also function as glucose sensors.

#### 2.1.1. Characteristics of Human Sodium-Independent Glucose Transporters, GLUT Proteins

Human sodium-independent glucose transporters (GLUT proteins, *SLC2A* genes, facilitating transport) are uniporters, i.e., integral membrane proteins involved in facilitated diffusion. These transporters transport a single water-solute molecule across its gradient, and may not utilize energy. To date, 14 members of the family of GLUTs have been identified (GLUT1–GLUT14, encoded by the *SLC2A1*–*SLC2A14* genes). One or more GLUT proteins are expressed in every type of cell; they possess various substrate specificities. These carrier proteins are involved in the transport of several hexoses, inositol, urate, glucosamine, and ascorbate [11]. All members of this family are facilitative transporters, except for GLUT13 (HMIT), which is an H^+^/*myo*-inositol symporter [12].

Human GLUT proteins contain 12 hydrophobic membranes spanning α-helical transmembrane (TM) domains, connected by a hydrophilic loop between TM6 and TM7 of GLUT [13]. These glucose transporters contain a single glycosylation site at the exofacial end, either between TM1 and TM2 (first extracellular loop) or between TM9 and TM10 (fifth extracellular loop). GLUTs also contain a short intracellular N-terminal segment, and a large C-terminal segment [11]. Based on the phylogenetic analysis of sequence similarity and characteristic elements, the family of GLUT1 protein is divided into three classes: Class I GLUT proteins includes GLUT1–GLUT4 and GLUT14; Class II comprises GLUT5, GLUT7, GLUT9 and GLUT11; Class III Gluts includes GLUT6, GLUT8, GLUT10, GLUT12, and GLUT13 (HMIT) [14]. Classes I and II contain an N-linked glycosylation site in the first exofacial loop between TM1 and TM2, and Class III contains the glycosylation site within the larger loop 9 [14].

The brain is protected by brain endothelial cells, which form a selective blood–brain barrier (BBB). This barrier restricts the passage of substances into and out of the brain. In the transport of substances, an important role is played by various transporters, expressed at the BBB on both the luminal (blood facing) and the abluminal (brain facing) surfaces of the neurovascular barrier. Ten GLUT proteins have been identified in the CNS [6,15,16,17], especially in the brain, as well as in peripheral nerves [18]. Some GLUTs are expressed in specific brain regions. A major role in the glucose uptake in the brain is played by GLUT1 and GLUT3.

GLUT1 is expressed in the human brain as a 45-kDa isoform and a 55-kDa isoform. The differences in their molecular weight are related to the differences in the N-linked glycosylation. GLUT1 55 kDa is present in the cerebral cortex, cerebral microvessels, luminal membrane in the BBB, cytosol and abluminal membrane of capillary endothelial cells. A less glycosylated isoform of GLUT1 is localized in glia (astrocytes and oligodendrocytes). Importantly, GLUT1 is not expressed in neurons [6,15,16].

GLUT2 is a poorly studied glucose transporter in the human brain [19]. It is expressed in astrocytes [20], especially in such brain areas as the hypothalamus, brain stem nuclei and tanycytes [21,22]. Low levels of GLUT2 mRNA were detected in the nucleus tractus solitarius, motor nucleus of the vagus, paraventricular hypothalamic nucleus, lateral hypothalamic area, arcuate nucleus and olfactory bulbs [23]. GLUT2 is also expressed in neurons [24]. It is suggested that GLUT2 may be involved in glucose sensing in the brain, which controls feeding, energy expenditure and counter regulation [25]. GLUT2 may also be involved in the regulation of neurotransmitter release and in glucose release by glial cells [24].

GLUT3, as mentioned earlier, is the predominant glucose transporter in the brain. It is expressed ubiquitously and abundantly in brain neurons [26]; therefore, it is referred to as a “neuronal glucose transporter.” In neurons, GLUT3 is detected in neurites, dendrites and cell bodies [27]. Its expression is also observed in brain microvessels, where it is localized to endothelial cells [28], and in brain astroglial cells [29]. It is suggested that insulin stimulates translocation of GLUT3 from the intracellular compartment into the plasma membrane, increasing glucose uptake by neurons [30].

GLUT4 is an insulin-sensitive glucose transporter. Results obtained in animal studies revealed its expression in the hypothalamus [31], cerebellum [32], sensomotor cortex [33], hippocampus [34] and pituitary gland [35]. Its physiological role in the brain remains unknown and needs further investigation.

GLUT5 is the main fructose transporter in the body. Its expression was detected in human microglial cells [36], BBB [37] and cerebral Purkinje cells in the fetus [38]. The physiological role of this glucose transporter in the microglia remains unclear. The level of fructose in the brain is low, and the transport activity for glucose is much lower than that of fructose [39]. Based on animal studies, it is suggested that fructose may pass the BBB, and brain cells may use fructose as a source of energy [40]. Rat neocortical cells may utilize fructose; however, this glucose transporter is expressed at only 4% of the level of the neuronal GLUT3 [41]. The physiological role of GLUT5 in the brain remains unknown and needs to be studied.

GLUT6 is a poorly studied GLUT protein in the human brain, and there is little information regarding its sites of expression and physiological role in CNS. It is suggested that its function involves transport of hexoses or related compounds across intracellular organelle membranes [11].

The GLUT8 protein is detected in human ependymal cells [42], whereas its mRNA is observed in the cerebellum, brainstem, hippocampus and hypothalamus [43]. Animal studies revealed the presence of GLUT8 in the hippocampus (in principal neurons, pyramidal and granule neurons and non-principal GABA neurons) [44], cortex, amygdala and the cerebellum [43]. GLUT8 may be involved in the transport of glucose out of the rough endoplasmic reticulum into the cytosol, causing impairment of glucose homeostasis in hippocampal neurons [45,46].

The expression of GLUT10 in the brain is only demonstrated using northern blot; therefore, its precise function and localization remains unknown. It is probably localized intracellularly [47].

GLUT12 is expressed in the human frontal cortex [48], but its precise function in brain remains unknown.

GLUT13 (HMIT) is an H^+^/*myo*-inositol cotransporter. It is predominantly expressed in the brain, with high abundance in the hippocampus, hypothalamus, cerebellum and brainstem. The presence of GLUT13 is detected in neurons and glial cells [11]. Its function may consist of the regulation of processes in the brain that require high levels of *myo*-inositol, which is a precursor for phosphatidylinositol, i.e., an important regulator of signaling pathways [49].

#### 2.1.2. Characteristics of Human Sodium-Dependent Glucose Transporters, SGLT Proteins

In humans, sodium-dependent glucose transporters are encoded by genes from the *SLC5A* family. To date, 12 genes in humans, *SLC5A1–SLC5A12*, have been described. Proteins encoded by these genes are cotransporters (SGLT proteins, secondary active transport), which transport sodium ions with different substances, such as glucose, galactose, mannose, fructose, *myo*-inositol, iodine, choline and water [10,16], except for SGLT3 (alias SAAT1), encoded by the *SLC5A4* gene, which has no glucose transport activity and plays a role of a glucose sensor [50]. The *SLC5A* genes encode proteins with 580–718 amino acid residues with a predicted mass of 60–80 kDa. Proteins encoded by the *SLC5A* genes contain 14 TM α-helices in total, although not in the sodium-iodide symporter (NIS) and SMCT1, both of which lack TM14 [51]. Both the hydrophilic N- and C-termini are located on the extracellular side of the cell membrane [10]. SGLTs are highly glycosylated membrane proteins and there are a variable number of consensus sites for N-linked glycosylation. In humans, there are 10 genes expressed in the CNS [16].

SGLT1 is expressed in neurons (hippocampal and cortical granule/pyramidal) [10,18], BBB [52] and endothelial cells of the luminal membrane of intracerebral capillaries [53]. In addition to two Na^+^ ions and one glucose molecule, this transporter can also transport 264 [54] or 249 [55] water molecules. It has a relatively minor role in the overall utilization of glucose in the brain [56]. Therefore, different roles in the brain have been suggested. When the level of glucose is decreased, SGLT1 is upregulated [57], which may be important for the arterial wall under hypoglycemia [58]. During ischemic conditions, it may play a role at the BBB in the transport of glucose from the inside of brain endothelial cells to the brain extracellular fluid [52]. During oxygen glucose deprivation, SGLT1 may transport Na^+^ ions and glucose into and out of cells when provided with a sodium-driving force [52]. It may also behave as a glucose receptor in the brain [54].

There is little information regarding the expression of SGLT2. Its expression was detected in the brain and in the BBB [10,18]. At a very low level, SGLT2 mRNA has been detected in the human cerebellum [59] and in isolated brain microvessels [60]. The functional role of SGLT2 in the brain has not been established. Its role in overall utilization of glucose in the brain is low. It may play a role similar to SGLT1 in pathological conditions [56]. This glucose cotransporter transports one Na^+^ ion for each glucose molecule.

Expression of SGLT3 is observed in human cholinergic neurons of the submucosal and myoenteric plexus, at the neuromuscular junction [10], and in the brain [10,61]. SGLT3 does not transport glucose but is a glucose sensor.

SGLT4 is expressed in the brain, but its further role remains unknown [10,61]. It is suggested to be involved in mannose homeostasis [62].

SGLT6, also known as SMIT2, is expressed in the brain (neurons) and spinal cord. Its mRNA is widely detected in the human brain: in the amygdala, caudate, cerebellum, dorsal raphe nucleus, hippocampus, hypothalamus anterior and posterior, locus coreuleus, medulla oblongata, nucleus accumbens and substantia nigra. In the brain cortex, SGLT6 mRNA is detected in the cingulate anterior and posterior, frontal lateral and medial, occipital, parietal and dorsal raphe nucleus [59]. It transports *myo*-inositol and D-chiro-inositol, regulating inositol levels in the brain [63]. The level of inositol in the brain is 100-fold greater than in the peripheral nervous system [64].

Studies related to the reminder of the sodium-dependent symporters in the brain are scarce. It has been found that SMVT, SMCT1 and SMCT2 are expressed in the brain, whereas SMIT1 is detected in the choroid plexus and the brain and CHT1 is expressed in the spinal cord and the medulla. However, their function remains unknown [10,18,61]. Therefore, their precise localization in the brain and functional role needs further investigations.

#### 2.1.3. Characteristics of Human SWEET Protein

Human SWEET1 is encoded by the *SLC50A1* gene. It transports mono- and disaccharides across the vacuolar and plasma membranes. SWEET1 has seven predicted TM domains with two internal triple-helix-bundles, counted by the translocation pathway (3 + 1 + 3 configuration) [10]. To date, human SWEET1 has not been detected in the human brain.

## 3. Changes in Glucose Metabolism in Alzheimer’s Disease

The brain accounts for 20% of an individual’s energy expenditure at rest [65]; however, it constitutes only 2% of the total body weight. Neurons in the brain utilize 70–80% of the total obtained energy, and the remaining portion is utilized by glial cells (astrocytes, oligodendrocytes and microglia) [66]. Decreased transport of glucose into the brain and decreased oxygen metabolic rates in the brain occur during normal aging, and may cause neurodegenerative disorders, such as Alzheimer’s, Parkinson’s and Huntington’s diseases [67]. PET imaging studies showed reduced glucose utilization in brain regions affected in patients with dementias [68].

In sporadic AD, the production of ATP from metabolism of glucose is decreased by 50%. The tendency to decrease is observed throughout the progression of the disease, whereby about 20% of energy deficit is observed [2,8]. Extensive synaptic loss is observed in advanced stages of AD. This pathology is associated with decreased uptake of glucose and glucose metabolism in specific brain areas [18,69,70]. Hypometabolism is an important risk factor in the development and progression of Alzheimer’s disease. There are suggestions that hypometabolism in AD may be caused by decreased glucose transport in the brain by glucose transporters and/or disturbances in brain metabolism [18].

There are also other hypotheses. Berlanga-Acosta et al. 2020 [71] suggested the nosogenic role of insulin resistance as a primary trigger with interrelated factors, such as neuroinflammation, the pathogenic role of Aβ and phosphorylated tau and mitochondrial dysfunction [71]. Pathologies such as Aβ42, aberrantly phosphorylated tau, white matter atrophy with loss of myelinated fibrils, neuroinflammation and loss of synapses could be explained by dysregulated insulin/IGF-1 signaling. As in the previous hypothesis, brain insulin resistance plays an important role in the development of these pathologies [72]. Brain insulin resistance and insulin dysregulation could contribute to the neurodegeneration observed in AD [73] and is an independent risk factor for cognitive impairment [74].

### 3.1. Expression of Glucose Transporters in Alzheimer’s Disease

The expression of glucose transporters is changed in the AD-affected human brain. Unfortunately, the literature on the pathophysiology of glucose transport in sporadic AD is very poor. The pathophysiology of these membrane carrier proteins underlying the disease is not well understood. Investigations of the brain of AD patients revealed decreased levels of GLUT1 and GLUT3 [75], especially in the cerebral cortex and hippocampus, with significant loss of GLUT3 [76,77,78]. A decreased level of GLUT3 in the cerebral cortex and hippocampus is due to downregulation of GLUT3 in neurons [75]. Decreased levels of GLUT1 and GLUT3 in the brain of AD patients may be caused by downregulation of hypoxia-inducible factor-1 (HIF-1), which is involved in the regulation of GLUT1 and GLUT3 [79] expression. This observation is suggested as a possible mechanism by which decreased levels of GLUT1 and GLUT3 causes neurodegeneration of the AD-affected brain. Researchers have also observed that, due to deficiency of GLUTs, glucose hypometabolism causes abnormal tau phosphorylation and/or neurofibrillary degeneration by downregulation of the hexosamine biosynthesis pathway. The decreased levels of GLUT1 and GLUT3 are correlated with a decrease in O-GlcNAcylation [79]. Another mechanism, related to the decrease of GLUT3 expression, has also been suggested [80]. According to this suggestion, the promoter of human GLUT3 contains three potential cAMP response element (CRE)-like elements: CRE1, CRE2 and CRE3. Significantly increased expression of GLUT3 may be related to overexpression of CRE-binding protein (CREB) or activation of cAMP-dependent protein kinase. In the AD brain, decreased full-length CREB was detected, whereas truncation of CREB was increased. Truncation is correlated with activation of calpain 1 in the human brain. Calpain 1 proteolyses CREB and generates a truncated CREB, which shows less activity to stimulate GLUT3 expression. In the human brain, GLUT3 correlates positively with full-length CREB and negatively with activation of calpain 1. Overactivation of calpain 1 caused by calcium overload proteolyses CREB, causing a decrease in GLUT3 expression, which impairs glucose uptake and metabolism in the AD-affected brain [80]. On the other hand, in the brain of AD patients, the level of GLUT2 is significantly increased, whereas no change is detected in the level of GLUT4 [79]. The level of GLUT2 in the AD-affected brain is more than two times higher than in healthy controls [79]. Researcher have observed that an increased level of GLUT2 is correlated to a similar extent with the activation of astrocyte; however, not only astrocytes but also tanycytes are expressed in the brain [24]. Therefore, it is suggested that the increased level of GLUT2 is related to activation of astrocytes in AD [79]. A significantly high level of GLUT12 was detected in the frontal cortex of AD patients [48]. Unfortunately, there is insufficient information on this issue.

### 3.2. Metabolic Disorders in Alzheimer’s Disease

One of the most prominent features of the underlying metabolic abnormalities in Alzheimer’s disease may be brain glucose metabolism. Disturbances in cerebral glucose metabolism are detected using the FDG PET technique. This is also one of the earliest pathologic events in this disease [81]. Decreased metabolism in the brain is first observed in the parietal-temporal area, posterior cingulate cortices and medial temporal lobes. It may progress to the frontal lobes, subcortical areas and the cerebellum [82]. On the other hand, brain metabolism abnormalities may be found in the earliest, pre-mild cognitive impairment (pre-MCI). Investigations of elderly healthy subjects revealed interesting results. Researchers have found that reduced FDG-PET brain metabolism predicts clinical progression in elderly healthy subjects [83]. Another study was performed on individuals with subjective memory impairment (SMI). The results obtained in FDG-PET and structural MRI methods showed hypometabolism in the SMI group in the precuneus and hypermetabolism in the right medial temporal lobe and reduced gray matter volume in the right hippocampus. Researchers observed that decline of longitudinal memory in these participants was associated with reduced glucose metabolism in the right precuneus [84]. Reduced of FDG-PET in the hippocampus, amygdala, posterior cingulate, superior parietal, entorhinal cortices, frontal cortex and inferior parietal region may reflect the pathophysiological changes in specific brain regions. These changes occur in preclinical Alzheimer’s disease [85]. Another study was performed on patients with SMI and MCI who underwent FDG-PET and detailed neuropsychological testing. Hypometabolism was observed in periventricular regions of patients with SMI, whereas in patients with MCI, hypometabolism was detected in the parietal, precentral and periventricular regions. Based on the obtained results, researchers suggest that hypometabolism in the periventricular regions, seen as reduced FDG-PET, may play a role as a predictive biomarker of pre-dementia and the extension of reduced glucose metabolism into parietal region may be associated with the progression of cognitive deterioration [86].

A dramatic decrease in the activities of enzymes, involved in glucose metabolism, such as phosphofructokinase (PFK), phosphoglycerate mutase, aldolase, glucose-6-phosphate isomerase and lactate dehydrogenase, was observed in AD-affected brains [87]. Additionally, decreased activities of the pyruvate dehydrogenase complex [88], cytochrome oxidase [89] and the α-ketoglutarate dehydrogenase complex were observed as well [90].

A study performed on animal models of AD revealed that decreased levels of Glut1 worsened amyloid pathology, neurodegeneration and cognitive function [91]. On the other hand, supplementation of ketone and nicotinamide was found to reduce Aβ and p-tau pathologies and improve behavioral outcomes [92,93].

As mentioned earlier, glucose is the primary fuel for the brain. During periods of prolonged fasting, the brain can use ketone bodies as the main source of energy. Decreased levels of glucose and insulin during prolonged fasting lead to the release of free fatty acids, which are used in the mitochondria in the beta-oxidation process. Excess acetyl-CoA stimulates production of ketone bodies, which there are extra acetyl-CoA and may be used in the Krebs cycle [94]. Therefore, it is suggested that ketone bodies can compensate for the energy deficit in patients with AD [94,95].

## 4. Alzheimer’s Disease as Type 3 Diabetes Mellitus

The connection between Alzheimer’s disease and diabetes was suggested a long time ago. For example, various results revealed that type 2 diabetes mellitus is a risk factor for Alzheimer’s diseases, and patients with AD have a higher risk to develop type 2 diabetes [96]. Several studies revealed links between Alzheimer’s disease and diabetes mellitus [17,18,96,97,98]. Therefore, an idea has been proposed that Alzheimer’s disease can be an “insulin-resistant state” or even a “type 3 diabetes” [99,100].

### 4.1. The Role of Insulin in the Central Nervous System

Insulin is a peptide hormone produced and secreted by pancreatic β-cells. The hormone can cross the blood–brain barrier from the circulation to the brain through a saturable receptor-mediated process [101]. There are also regions in the brain, e.g., the hypothalamus and choroid plexus, that may serve as a more rapid site of entry of peripheral insulin into the CNS [102]. It is also suggested that insulin is synthesized in the brain. Its synthesis may occur in pyramidal neurons from the hippocampus, prefrontal cortex, entorhinal cortex and olfactory bulb [103,104]. This hypothesis is based on results obtained in animal studies. Preproinsulin I and II mRNA was detected in the rat fetal brain [105], and the presence of C-peptide in cerebrospinal fluid was reported in a clinical study [106].

Insulin and insulin-like growth factor 1 and 2 (IGF-1 and IGF-2) play an important role in brain function. Neurons and glia highly express insulin and IGF receptors [107]. Insulin and IGFs are involved in several processes in the brain, such as the metabolism of glucose and lipids and the regulation of neuronal development, memory, learning and cognitive functions [107,108]. These polypeptides participate in neuronal and glial functions, such as growth, survival, expression of genes, cytoskeletal assembly, neurotransmitter function and synapse formation [109,110]. Insulin regulates the concentration of several neurotransmitters, e.g., acetylcholine, norepinephrine and nephrine, which have essential roles in the formation of memory [111]. This hormone supports neuronal plasticity and cholinergic functions necessary for learning, memory and myelin maintenance [112]. Insulin binds to the insulin receptor (IR). In the adult nervous system, insulin binds to isoform IR-A, whereas isoform IR-B is found mainly in adipose tissue, liver and skeletal muscle [113]. IGF-1 and IGF-2 bind to IGF-Rs; however, these growth factors can bind to the IR. Insulin receptors are tyrosine kinases, containing α and β subunits. Once insulin or IGFs bind to the IR, the α subunit promotes autophosphorylation of tyrosine residues, which are present in the β subunit, causing phosphorylation of insulin receptor substrates 1 and 2 (IRS-1 and IRS-2). Phosphorylated IRSs may activate two parallel functional signal transduction cascades: the phosphatidylinositol-3 kinase (PI3-K), and mitogen-activation protein kinase (MAPK) pathways. The activation of the PI3-K pathway subsequently activates protein kinase B (PKB), also known as AKT. The activation of this pathway stimulates translocation of GLUT4 from the intracellular compartment into the cell membrane and, consequently, increases cellular glucose uptake. Activation of this pathway also triggers a large variety of biological actions, such as synthesis of protein and glycogen as well as anti-lipolytic and anti-apoptotic activities. It plays an important role in cell proliferation and insulin action in cells and is involved in the biosynthesis of GLUT1. In turn, activation of the MAPK pathway regulates the expression of genes associated with glucose metabolism and induces mitosis in cells. This signal is involved in the biosynthesis of GLUT3 [7,15,16,19,97,98,114]. These observations suggest that insulin signaling is an important pathway in the brain.

### 4.2. Impairment of the Brain Insulin/IGF Signaling Pathway

Several studies suggest a role of insulin dysfunction in the pathogenesis of sporadic Alzheimer’s disease. A deficit in insulin/IGF signaling begins early and progresses with the development of the disease. A deficit in this signaling is related to the effect of insulin resistance and deficiency [5]. Reduced levels of insulin and insulin receptors have been found in AD brains [109]. In patients with AD without diabetes, the responses to insulin/IGF-1 signaling in the PI3-K signaling pathway are markedly reduced. Reduced insulin/IGF-1 responses are maximal at the level of IRS-1 [115].

The activation of the PI3-K signaling pathway stimulates translocation of GLUT4, increasing cellular glucose uptake. Impairment in GLUT4 functions in the brain caused by insulin/IGF resistance reduces glucose uptake and utilization. This pathology compromises cell energy and homeostatic functions and disrupts neuronal skeleton and synaptic connection [116].

Hypometabolism increases oxidative and endoplasmic reticulum stress and mitochondrial dysfunction and generates reactive oxygen (ROS) and reactive nitrogen (RNS) species [117]. Increased oxidative stress, ROS and RNS damage DNA, RNA and proteins. Increased production of lipid peroxidation, deficits of energy and cell death are observed as well as increased expression of AβPP, deposition of Aβ42 and fibrillarization [116]. Insulin/IGF resistance in the AD brain stimulates pro-inflammatory and pro-death cascades and downregulates the expression of genes involved in cholinergic homeostasis [5,118]. Investigations performed on AD patients revealed a correlation between insulin resistance and increased levels of Aβ and amylin (islet amyloid polypeptide, IAPP) [119,120]. Animal studies have shown that insulin deficiency in the brain may stimulate the formation of Aβ due to upregulation of amyloid precursor protein (APP) and beta-secretase 1 (BACE-1), which are involved in Aβ formation [121]. Insulin and IGF-1 inhibit the synthesis of Aβ, regulating the glycogen synthase kinase-3β (GSK-3β) activity [122]. Insulin and IGF increase the extracellular secretion of Aβ and APP in the brain and can accelerate its trafficking from the trans-Golgi network to the plasma membrane [123]. This process is mediated by the MAPK signaling pathway, which is insulin-dependent. Under chronic peripheral hyperinsulinemia, this process does not occur. The initially high level of insulin and insulin in the brain decreases, because its synthesis and/or transport in the brain are downregulated [124]. An important role in the degradation of Aβ is played by the insulin-degrading enzyme (IDE), which is highly expressed in the brain. This metalloprotease catabolizes insulin and IGF-1 and may degrade Aβ [124,125]. This process occurs through a PI3-K-dependent mechanism [126]. Therefore, insulin resistance may inhibit IDE [109]. Investigations performed on mammalian models of AD revealed that overexpression of Aβ or induction by an intracerebral injection causes deficiency of the insulin signal [127,128]. Aβ can inhibit insulin binding to the insulin receptor, causing loss of the insulin signal [129].

Stimulation of insulin/IGF may regulate the expression of the *tau* gene and phosphorylation of proteins [130,131]. Kinases such as GSK-3β, intracellular-regulated kinase 1/2 (ERK1/2), and cyclin-dependent kinase 5 (Cdk-5), activated by insulin/IGF-1, are involved in the process of tau phosphorylation [109,132]. Insulin/IGF-1 resistance in the brain decreases this signaling, causing increased activation of GSK-3β [133]. Overactivation of GSK-3β causes hyperphosphorylation of tau. Hyperphosphorylated tau proteins bind to each other and tie themselves into “knots.” These insoluble aggregates, known as intracellular neurofibrillary tangles (NFTs) are accumulated in cell bodies and proximal dendrites, destabilizing microtubules involved in microtubule-dependent processes [134], such as exacerbation of cytoskeletal collapse, neurite retraction and synaptic disconnection [116]. NFTs also promote oxidative stress, apoptotic or necrotic death and mitochondrial dysfunction [109]. Negative regulation of tau protein phosphorylation due to insulin was observed in cultured human neuronal cells [135]. Insulin and IGF-1 may inhibit hyperphosphorylation of tau protein by the stimulation of Akt-induced phosphorylation/inactivation of GSK-3β [109].

## 5. Brain Glucose Transporters as Targets for Alzheimer’s Disease

Alzheimer’s disease is associated with hypometabolism due to decreased glucose transport into brain cells. Decreased glucose uptake may be caused by several factors, e.g., downregulation of glucose transporters in neurons, impaired glycolysis, disturbances in brain mitochondria, insulin/IGF resistance, changes in intracellular signaling pathways and so on. Therefore, increased glucose transport into neurons and/or glial cells may be a therapeutic approach in AD.

### 5.1. Role of Brain Glucose Transporters

As mentioned above, Alzheimer’s disease may be regarded as a metabolic disease, i.e., type 3 diabetes mellitus. Therefore, it is suggested that antidiabetic drugs may be administered in AD. There are also other actions proposed in AD treatment (Table 1).

#### 5.1.1. Antidiabetic Drugs in AD

##### Metformin

Metformin is the gold-standard drug for the treatment of type 2 diabetes. It stimulates glucose uptake by cells. Animal studies revealed that restored glucose levels in the brain may prevent the progression of AD. The effectiveness of increasing glucose uptake in neurons was investigated using a genetic approach in a *Drosophila* model of Alzheimer’s disease. Researchers selectively overexpressed GLUT1 in adult neurons of the transgenic fly, which express a mutant form of Aβ. Overexpression of this glucose transporter in the brain of the transgenic fly increased the lifespan and ameliorated the cardinal features of AD [136]. Similar results were obtained after treatment of *Drosophila* with metformin. Administered metformin in the AD fly model also delayed the progression of experimental AD in *Drosophila.* In animal models, metformin increased the lifespan and rescued deficits of experimental AD. These results were consistent with previous results obtained in mouse models of AD [127]. Based on the results, it was suggested that increased glucose uptake in the brain may also reduce protein aggregation in AD by restoring proteostasis [137]. Protective effects were also observed even after the neuropathological process had begun. This observation suggests that increased glucose uptake in the brain may be a therapeutic potential in AD [138].

##### Inhibitors of SGLT2

SGLT2 inhibitors are a class of oral anti-hyperglycemic agents suggested for the treatment of diabetes mellitus [139]. These inhibitors decrease glucose levels independently from insulin by decreasing tubular glucose reabsorption. Among inhibitors of SGLT2, empagliflozin (EMP) has been comprehensively investigated. In patients with type 2 diabetes, this inhibitor was found to control hyperglycemia and reduces cardiovascular comorbidities and deaths associated with T2DM. Animal studies performed on a mixed murine model of AD and T2DM revealed reduction of AD complications after treatment with EMP [140]. AD-T2DM mice (APP/PS1xdb/db) were treated with EMP 10 mg/kg for 22 weeks. The treatment with EMP in these mice improved of brain atrophy by reduction of neuronal loss, significantly reduced hyperglycemia and reduced the burden of spontaneous hemorrhages. Researchers also observed decreased levels of soluble and insoluble Aβ40 and Aβ42 in the cortex and hippocampus and reduction of tau phosphorylation in these brain regions. The inhibitor improved learning and memory in the AD-T2DM mice, as observed in the Morris water maze. Insulin levels were improved as well. Based on these results, the researchers postulated a beneficial role of EMP in reduction of AD complications in the brain [140].

##### GLP-1 Agonist

Glucagon-like peptide-1 (GLP-1) is an incretin hormone. It was observed that native GLP-1 increases the maximum glucose transport capacity in brain capillary endothelium [141,142]. In the therapy of neurodegenerative diseases, such as AD, analogs of GLP-1 may be used [143]. GLP-1 prevents the decline in the cerebral metabolic rate for glucose (CMR_glc_) in AD. It may raise the numbers of glucose transporters in the blood–brain barrier. These compounds are well established as effective agents for the treatment of patients with T2DM. Therefore, it was suggested that analogs of GLP-1 may have beneficial effects in patients with AD. Investigations were performed with liraglutide, an agonist of GLP-1 [144]. Human studies were performed on patients with AD treated with liraglutide or placebo for 6 months. A significant increase in the blood–brain glucose transfer capacity (*Tmax*) estimates of cerebral cortex, equal to *Tmax* estimates in healthy subjects, was demonstrated in AD patients treated with liraglutide compared to those treated with the placebo. The researchers also observed that the *Tmax* declined with the duration of AD, and the agonist of GLP-1 enhanced the activity of GLUT1 in the BBB. Therefore, the authors conclude that treatment with a GLP-1 analog reduces the effects of disease duration [144]. They suggest that the increased *Tmax* may be associated with an increase in the levels of GLUTs or increased postprandial insulin levels. Astrocytic insulin receptors influence GLUT1 expression and, consequently, the levels of this glucose transporter in the plasma membrane. Animal studies performed on a murine model of diabetes showed that combination of liraglutide with dapagliflozin, i.e., a SGLT2 inhibitor, had beneficial metabolic and neuroprotective effects. Administration of liraglutide + dapagliflozin significantly improved learning and memory in diabetic mice. Histological changes were also observed in the brain. Increased numbers of immature neurons in dental gyrus and synaptic density in the stratum oriens and stratum pyramidale were detected [145].

##### Insulin

Insulin may be considered as a therapeutic agent against AD. The absence or decreased levels of insulin may cause desensitization of *N*-methyl-D-aspartate (NMDA)-dependent processes, impairment of learning and deaths of neurons. Inhibition of GSK-3β by PI3-K/Akt signaling stimulates phosphorylation of tau, accelerating progression of AD. Therefore, it is suggested that increased levels of insulin may alleviate the symptoms of AD. It was observed that intracerebroventricular administration of insulin improves memory performance. However, this approach raises questions concerning the applicability of insulin to AD patients [146]. Intranasal administration of insulin via transport along olfactory and trigeminal perivascular channels through axonal transport pathways has been studied [96]. The use of insulin as a therapeutic agent in AD was also suggested in earlier research. It was found that increased levels of insulin enhanced memory in adults with AD. The researchers also found that somatostatin, but not glucose, enhanced memory [147]. The role of insulin as a therapeutic agent in AD has also been suggested by other authors [148].

#### 5.1.2. Other Compounds in Therapy of AD

D-galactose is a C-4 epimer of glucose. In the human and animal body, it is found in very small quantities. At high levels, it reacts with the free amines of amino acids in proteins and peptides. This reaction forms advanced glycation end products, which cause oxidative damage in the body. There are a few studies on the effects of oral galactose on the brain. For example, oral galactose was found to increase the activity of the mitochondrial respiratory chain complex, in the prefrontal cortex and hippocampus in animal models. Galactose enters the enterocytes by SGLT1 and is transported out of enterocytes by GLUT2 at the basolateral membrane into the circulation. From circulation, it is transported into neurons in the brain by GLUT3. As revealed in animal studies, oral administration of galactose increases the expression of hippocampal GLUT3 [149]. Noteworthy, GLUT3 is the main glucose transporter in the brain, and its expression in AD is decreased. In cells, galactose is metabolized to glucose. Therefore, when the intracellular glucose level is decreased, as in sporadic Alzheimer’s disease, galactose may be an alternative source of energy in neurons. Galactose may also have other beneficial effects on the AD brain. In the brain, it is metabolized into amino acids, increasing the level of glutamate and GABA, which play a role in normal cognitive functioning, increase the intracellular concentration of galactose or its derivatives, and may normalize O-GlcNAcylation of the regulatory proteins, as in the case of tau protein [150].

Curcumin is an active compound extracted from the root of the herb *Curcuma longa.* Its therapeutic potential in diseases such as cancer, diabetes, autoimmune and neurodegenerative diseases has been demonstrated in a number of studies [151,152]. Animal studies using an APPswe/PS1dE9 double transgenic mouse model have revealed that administration of curcumin significantly increases Aβ- and insulin-degrading enzymes and significantly decreases Aβ40, Aβ42 and aggregation of Aβ-derived diffusible ligand (ADDL) expression in the hippocampus. It was also observed that spatial learning and memory ability were improved in these mice [153]. In another study on these double transgenic mice, researchers observed that the administration of curcumin significantly improved the expression of Glut1 and Glut3 in the brain of AD mice. High glucose cerebral metabolism was found in these mice. The authors conclude that curcumin ameliorates the impaired insulin signaling in the IR/IRS-1/PI3-K/Akt and IGF-1R/IRS-2/PI3-K/Akt pathways [154].

LP9M80-H, i.e., a compound from the herb *Liriope platyphyla*, which regulates the biosynthesis of Glut1 and Glut3 in ICR mice, has been suggested for the treatment of Alzheimer’s disease as well [114].

### 5.2. Other Suggested Therapeutics for AD Treatment

#### 5.2.1. Small Interfering RNAs

Small interfering RNAs (siRNAs) show great potential for Alzheimer’s disease therapy. These molecules act by specific silencing of BACE1 (β-site APP cleavage enzyme 1), which ameliorates the neuropathology of AD. The delivery of BACE1 siRNA (siBACE1) to the mouse brain via systemic injection may partially reduce the neuropathology of AD [155]. Unfortunately, the therapeutic effect was not ideal, probably due to the low accumulation of siRNA in the brain and its stability. Therefore, researchers have developed a glycosylated “triple-interaction” stabilized polymeric siRNA nanomedicine (Gal-NP@siRNA). It shows superior blood stability and can penetrate the BBB via Glut1 [156]. Its administration in an APP/PS1 transgenic AD mouse model was reported to ameliorate AD-like pathology and restore the deterioration of cognitive capacity in these mice without side effects. The authors conclude that this therapy may be effective in the treatment of neurodegenerative diseases [156].

#### 5.2.2. Liposomes

The brain-derived neurotrophic factor (BDNF) plays an important role in the development, maintenance and plasticity of the central nervous system. Its level in AD patients is significantly reduced, causing reduced plasticity and neuronal death. This pathology is associated with neurodegeneration in the hippocampus and cortex of AD patients. Therefore, it is suggested that delivery of BDNF to the brain may potentially ameliorate AD pathology. Researchers described the targeted delivery of the BDNF gene to the brain. Modified liposome nanoparticles were used in this experiment [157]. The surface of liposomes used in the experiments was modified with glucose transporter-1 targeting ligand (mannose) and cell-penetrating peptides (penetratin or rabies virus glycoprotein). These modifications caused selective and enhanced delivery to the brain, and showed significantly higher transfection of BDNF in primary astrocytes and neurons. The authors suggest that this strategy to increase BDNF protein levels in the brain may reverse AD pathophysiology [157].

#### 5.2.3. Regular Exercise

Lack of physical activity is predisposing factor for neurodegenerative disorders, such as Alzheimer’s disease and Parkinson’s disease, whereas physical exercises boost cognitive function [158], attention processing, speed memory and learning [67]. Exercises targets specific brain areas, such as prefrontal and medial temporal cortices [159] and the hippocampus [160]. Elderly people that regularly exercise have increased brain volumes in these network areas [161]. Regular exercise increases the expression of various proteins, for example, antioxidant enzymes, antiapoptotic proteins, proteins involved in mitochondrial biogenesis, protein chaperones, neurotrophic factors and fibroblast growth factor 2 [158,162]. Importantly, neurotrophic factor BDNF increases the expression of GLUT3, causing stimulation of energy metabolism. Exercise also triggers the intracellular signaling pathway, which is similar to the signaling pathway triggered by insulin, causing, e.g., translocation of GLUT4 from the intracellular compartment into the plasma membrane. An animal study has revealed that regular exercise decreases the expression of Aβ and phosphorylated tau production, with increased production of ATP in the brain. Researchers have found an increased number of synapses and improved expression levels of Glut1 and Glut3 expression in the central nervous system in a mouse model of AD [163].

## 6. Summary

Alzheimer’s disease is the most common cause of dementia. There are several factors increasing the risk of AD. One of these is impairment of brain energy. Hypometabolism is observed in the AD-affected brain due to decreased uptake of glucose and its metabolism. Reduced glucose uptake is related to decreased expression of glucose transporters in neurons, namely GLUT1 and GLUT3. Therefore, it is suggested that increased transport of glucose into neurons may be a therapeutic approach in AD. Several strategies have been proposed, such as administration of antidiabetic drugs, inhibitors of SGLT2 and agonists of GLP-1. The possibility of use of siRNA and liposomes has been investigated as well. A therapeutic effect is also achieved after regular exercise.

## Figures and Tables

**Table 1 ijms-22-08142-t001:** The effects of different therapies in Alzheimer’s disease.

Therapeutic Agent	Therapeutic Effects
Metformin	In animal models, metformin increased the lifespan and rescued deficits of experimental AD lifespan, ameliorates the cardinal features of AD and reduces protein aggregation in the disease by restoring proteostasis. It increases glucose uptake in the brain.
Inhibitors of SGLT2	The treatment with empagliflozin in AD-T2DM mice improved brain atrophy by a reduction of neuronal loss, which significantly reduced hyperglycemia and reduced the burden of spontaneous hemorrhages. EMP also decreased levels of soluble and insoluble Aβ40 and Aβ42 in the cortex and hippocampus and reduction of tau phosphorylation in these brain regions. The inhibitor improved learning and memory in the AD-T2DM mice, as observed in the Morris water maze.
Agonists of GLP-1	Human studies performed on patients with AD treated with liraglutide for 6 months revealed in these patients a significant increase in the blood–brain glucose transfer capacity (*Tmax*) estimates of the cerebral cortex, equal to *Tmax* estimates in healthy subjects. The researchers also observed that the *Tmax* declined with the duration of AD, and the agonist of GLP-1 enhanced the activity of GLUT1 in the BBB. Treatment with a GLP-1 analog reduces the effects of disease duration. The increased *Tmax* may be associated with an increase in the levels of GLUTs or increased postprandial insulin levels. Animal studies revealed that the use of this agonist increases the number of immature neurons in dental gyrus and the synaptic density in the stratum pyramidale.
Insulin	Intracerebroventricular administration of insulin improves memory performance. Increased levels of insulin enhances memory in adults patients with AD.
D-galactose	Oral galactose was found to increase the activity of the mitochondrial respiratory chain complex in the prefrontal cortex and hippocampus in animal models. As revealed in animal studies, oral administration of galactose increases the expression of hippocampal GLUT3. In cells, galactose is metabolized to glucose. Therefore, when the intracellular glucose level is decreased, galactose may be an alternative source of energy in neurons. In the brain, galactose is metabolized into amino acids, increasing the level of glutamate and GABA, which play a role in normal cognitive functioning, increase the intracellular concentration of galactose or its derivatives, and may normalize O-GlcNAcylation of the regulatory proteins, as in the case of tau protein.
Curcumin	Administration of curcumin significantly increases Aβ- and insulin-degrading enzymes, and significantly decreases Aβ40, Aβ42 and aggregation of Aβ-derived diffusible ligand (ADDL) expression in the hippocampus. Spatial learning and memory ability were improved in animal model of AD. Administration of curcumin significantly improved the expression of Glut1 and Glut3 in the brain of AD mice. High glucose cerebral metabolism was found in these mice. Curcumin ameliorates the impaired insulin signaling in the IR/IRS-1/PI3-K/Akt and IGF-1R/IRS-2/PI3-K/Akt pathways.
Small interfering RNAs	Administration of Gal-NP@siRNA, a glycosylated “triple-interaction,” stabilized polymeric siRNA in a APP/PS1 transgenic AD mouse model and was reported to ameliorate AD-like pathology and restore detoriation of cognitive capacity in these mice without side effects.
Liposomes	The levels of BDNF in AD patients is significantly reduced; therefore, it is suggested that delivery of BDNF to the brain may potentially ameliorate AD pathology. Researchers described the targeted delivery of the BDNF gene to the brain using modified liposome nanoparticle. The surface of liposomes used in the experiments was modified with glucose transporter-1 targeting ligand (mannose) and cell penetrating peptides (penetratin or rabies virus glycoprotein). These modifications caused selective and enhanced delivery to the brain, and showed significantly higher transfection of BDNF in primary astrocytes and neurons. This strategy to increase BDNF protein levels in the brain may reverse AD pathophysiology.
Regular exercise	Regular exercise increases the expression of various proteins, for example, antioxidant enzymes, antiapoptotic proteins, proteins involved in mitochondrial biogenesis, protein chaperones, neurotrophic factors and fibroblast growth factor 2. Importantly, neurotrophic factor BDNF increases the expression of GLUT3, causing stimulation of energy metabolism. Exercise also triggers the intracellular signaling pathway, which is similar to the signaling pathway triggered by insulin, causing, e.g., translocation of GLUT4 from the intracellular compartment into the plasma membrane. An animal study has revealed that regular exercise decreases the expression of Aβ and phosphorylated tau production, with increased production of ATP in the brain. Researchers have found an increased number of synapses and improved expression levels of Glut1 and Glut3 expression in the central nervous system in a mouse model of AD.

## Data Availability

Not applicable.

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
