# Peer review of "Brain Glucose Transporters: Role in Pathogenesis and Potential Targets for the Treatment of Alzheimer’s Disease"

_ijms, 2021, doi:10.3390/ijms22158142_

Round 1

Reviewer 1 Report

This a useful review of possible mechanisms that reagents might exploit in Alzhei

mer’s disease treatment. It needs a review by someoneGirl herbs and three Elysianby someone fluent in EnglishThere are numerous many areas that need sharpening of language to make the paper more readable.The author is urged to do this. 

Author Response

Dear Reviewer

Thank you very much for your opinion and suggestions. According to your suggestion, the language and style of the manuscript were checked and improved by the editing service. I hope that after proofreading, article will be more friendly for readers.

Reviewer 2 Report

The topic of this review article is very interesting and important, however the manuscript requires significant revisions. Description of the different human glucose transporters (Section 2.) is unnecessarily long, as this topic was summarized recently in detail in another paper of the author. Therefore I suggest to shorten and rework of this section, to avoid overlaps. Moreover figures/tables summarizing the main conclusions of the article would be very useful. Finally the language and style of the manuscript should be improved.

Author Response

Dear Reviewer

Thank you very much for your opinion and suggestions. According to your suggestion, Section 2 on human glucose transporters was shortened, however not extensive. The remaining informations describe localization of glucose transporters in CNS, and, in the case of some transporters, their role in the brain or glia. Also, according to your suggestion, there is added 1 table, which summarizes described methods of therapy. And the last suggestion, the language and style of the manuscript were improved by editing service. I hope that after introduced changes, manuscript will be more friendly for readers. Thank you very much.

Reviewer 3 Report

I reviewed the manuscript entitled "Brain glucose transporters. Role in pathogenesis and potential targets for the treatment of Alzheimer's disease".

The present manuscript focuses on an important field about pathogenetic mechanisms of AD, specifically alterations in the glucose metabolism and insuline-resistance, and subsequent oxidative stress and neurodegeneration. At last, therapeutic implications are very significant.

Some points should be improved.

1) 3. Changes of glucose metabolism in Alzheimer's disease.

"PET imaging studies showed reduced glucose utilization in brain regions affected in patients with mental diseases." Author refers to a paper focusing on dementias (AD and VaD); so, the term "mental diseases" is inappropriate and should be replaced. It is important to distinguish between neurologic pathologies (i.e., dementias) and neuropsychiatric diseases (i.e., schizophrenia) in which an alteration in glucose metabolism is also documented.  This point might be discussed.

2) "There are suggestions that hypometabolism in AD may be due to decreased glucose transport in brain by glucose transporters, and/or disturbances in brain's metabolism." This point should be further discussed. In fact, the author underlines in the manuscript the suggestive hypothesis that brain glucose disregulation may lead to neurodegeneration, also supported by several evidences. Below in the text it is reported that "decreased levels of GLUT1 and GLUT3 causes the neurodegeneration of AD-affected brain." However, this esplicative hypothesis could be accompanied by alternative one, in which the synaptic loss due to phosphotau-induced neurodegeneration may cause glucose imbalance.

I suggest  some recent papers to further discuss the keypoint:

Kellar D, Craft S. Brain insulin resistance in Alzheimer's disease and related disorders: mechanisms and therapeutic approaches. Lancet Neurol. 2020 Sep;19(9):758-766. doi: 10.1016/S1474-4422(20)30231-3;  Berlanga-Acosta J, Guillén-Nieto G, Rodríguez-Rodríguez N, Bringas-Vega ML, García-Del-Barco-Herrera D, Berlanga-Saez JO, García-Ojalvo A, Valdés-Sosa MJ, Valdés-Sosa PA. Insulin Resistance at the Crossroad of Alzheimer Disease Pathology: A Review. Front Endocrinol (Lausanne). 2020 Nov 5;11:560375. doi: 10.3389/fendo.2020.560375; de la Monte SM. Insulin Resistance and Neurodegeneration: Progress Towards the Development of New Therapeutics for Alzheimer's Disease. Drugs. 2017 Jan;77(1):47-65. doi: 10.1007/s40265-016-0674-0.; Ma L, Wang J, Li Y. Insulin resistance and cognitive dysfunction. Clin Chim Acta. 2015 Apr 15;444:18-23. doi: 10.1016/j.cca.2015.01.027. Epub 2015 Feb 4. 

3) Authors introduce the evidence that brain metabolism abnormalities can be found early in AD pathology. To this purpose, some papers report findings of this feature in the earliest, pre-MCI phases of AD (Subjective Cognitive Decline or cognitively normal individuals): 

Ewers, M.; Brendel, M.; Rizk-Jackson, A.; Rominger, A.; Bartenstein, P.; Schu , N.;Weiner, M.W.; Alzheimer’s Disease Neuroimaging Initiative (ADNI). Reduced FDG-PET brain metabolism and executive function predict clinical progression in elderly healthy subjects. NeuroImage Clin. 2014, 4, 45–52; Scheef, L.; Spottke, A.; Daerr, M.; Joe, A.; Striepens, N.; Kölsch, H.; Popp, J.; Daamen, M.; Gorris, D.; Heneka, M.T.; et al. Glucose metabolism, gray matter structure, and memory decline in subjective memory impairment. Neurology 2012, 79, 1332–1339; Shah, T.; Lenzo, N.; Salvado, O.; Taddei, K.; Verdile, G.; Gardener, S.L.; Sohrabi, H.R.; Shen, K.-K.; Rainey-Smith, S.R.; Weinborn, M.; et al. Cerebral Glucose Metabolism is Associated with Verbal but not Visual Memory Performance in Community-Dwelling Older Adults. J. Alzheimer Dis. 2016, 52, 661–672; Song, I.U.; Choi, E.; Oh, J.K.; Chung, Y.A.; Chung, S.W. Alteration patterns of brain glucose metabolism: Comparisons of healthy controls, subjective memory impairment and mild cognitive impairment. Acta Radiol. 2016, 57, 90–97; Hohman, T.J.; Beason-Held, L.L.; Lamar, M.; Resnick, S.M. Subjective Cognitive Complaints and Longitudinal Changes in Memory and Brain Function. Neuropsychology 2011, 25, 125–130.

Author Response

Dear Reviewer

Thank you very much for your opinion and suggestions.

1) According to Reviewer suggestion, term "mental disease" was changed on "dementias".

2) Point 2 was further discussed. And, according to Reviewer suggestion, the discussed points were based on literature proposed by Reviewer.

3) Similar as point 2, point 3 was discussed according to suggested literature by Reviewer.

Literature proposed in points 2 and 3 is very important in this manuscript. Article is more rich.

4) And the last suggestion, the language and style of the manuscript were improved by editing service.

Dear Editor, thank you very much for your opinion and suggestions. I hope, that introduce suggested changes cause manuscript more friendly for readers.

Round 2

Reviewer 2 Report

Thanks for the detailed responses.

Reviewer 3 Report

Please check the writing of "precuneus"